# Immediate Effects of Instrument-Assisted Soft Tissue Mobilization on Hydration Content in Lumbar Myofascial Tissues: A Quasi-Experiment

**DOI:** 10.3390/jcm12031009

**Published:** 2023-01-28

**Authors:** Andreas Brandl, Christoph Egner, Monique Schwarze, Rüdiger Reer, Tobias Schmidt, Robert Schleip

**Affiliations:** 1Department of Sports Medicine, Faculty for Psychology and Human Movement Science, Institute for Human Movement Science, University of Hamburg, 20148 Hamburg, Germany; 2Department for Medical Professions, Diploma Hochschule, 37242 Bad Sooden-Allendorf, Germany; 3Osteopathic Research Institute, Osteopathie Schule Deutschland, 22297 Hamburg, Germany; 4Institute of Interdisciplinary Exercise Science and Sports Medicine, MSH Medical School Hamburg, 20457 Hamburg, Germany; 5Conservative and Rehabilitative Orthopedics, Department of Sport and Health Sciences, Technical University of Munich, 80333 Munich, Germany

**Keywords:** instrument-assisted soft tissue mobilization, bioimpedance analysis, thoracolumbar fascia, water content

## Abstract

Background: Instrument-assisted soft tissue mobilization (IASTM) is thought to alter fluid dynamics in human soft tissue. The aim of this study was to investigate the influence of IASTM on the thoracolumbar fascia (TLF) on the water content of the lumbar myofascial tissue. Methods: In total, 21 healthy volunteers were treated with IASTM. Before and after the procedure and 5 and 10 min later, lumbar bioimpedance was measured by bioimpedance analysis (BIA) and TLF stiffness was measured by indentometry. Tissue temperature was recorded at the measurement time points using an infrared thermometer. Results: Bioimpedance increased significantly from 58.3 to 60.4 Ω (*p* < 0.001) at 10-min follow-up after the treatment. Temperature increased significantly from 36.3 to 36.6 °C from 5 to 10 min after treatment (*p* = 0.029), while lumbar myofascial stiffness did not change significantly (*p* = 0.84). Conclusions: After the IASTM intervention, there was a significant increase in bioimpedance, which was likely due to a decrease in water content in myofascial lumbar tissue. Further studies in a randomized control trial design are needed to extrapolate the results in healthy subjects to a symptomatic population as well and to confirm the reliability of BIA in myofascial tissue.

## 1. Introduction

There is increasing evidence that the thoracolumbar fascia, with its force transmission properties and sensory innervation, is directly related to nonspecific low back pain [1,2,3,4,5,6,7,8]. Specifically, the multilayered thoracolumbar fascia (TLF) consists of dense aponeuroses that attach to the posterior processes of the vertebrae as well as the supraspinal ligaments in the thoracic and lumbar spine. It is these aponeuroses that can withstand loads [1] and transmit forces [9]. Beyond that, they have thin layers of loose connective tissue that separate these aponeuroses as well as the underlying epimysial muscle layers and support shear mobility, which plays an important role in the movement mechanics of the trunk [10].

Each individual layer of the TLF is connected to the immediately adjacent one by a thin layer of loose connective tissue, which is rich in ground substance (GS) (i.e., water, glycosaminoglycans including hyaluronic acid, proteoglycans and glycoproteins) [11]. Fasciacytes, fibroblast-like cells located along the surface of the fibrous fascia layers, synthesize and secrete hyaluronic acid [12]. Hydration dynamics are critical to the ability of fascial layers to slide against each other [13]. This is particularly important in light of the findings of Langevin et al. [2] that people with chronic low back pain have significantly lower TLF mobility. If the fluid in the extracellular matrix cannot escape due to the impermeability of the TLF and the venous outflow channels collapse, a compartment syndrome may occur in which the intrafascial pressure exceeds the venous pressure [14]. On the other hand, plasma injected into the fascial layers has been observed to separate the fascial layers and increase their separation [15]. Therefore, many manual therapy techniques target this mechanism to restore fluid homeostasis and the ability of fascial layers to glide on each other. Roman et al. [15] built a mathematical model in which they analyzed hyaluronic acid around fascia under different pressure modalities (including frequency, force vector) as a result of manual therapy and found that fluid pressure increased dramatically when fascia was deformed during treatment. This separated the adjacent fascia layers, creating a thicker GS interstitial space that could improve gliding and, consequently, muscle work. Recent studies indicate an interaction between the viscosity of the GS, especially hyaluronic acid, the speed of movement and the hardness of the fascia [13]. Roman et al. [15] suggest that instrument-assisted soft-tissue manipulation (IASTM) with hand-held devices such as the Graston technique applies pressure to a much smaller area compared to the therapist’s fingers or hand and therefore may provide benefits through a greater pressure gradient between fascial layers with the GS. However, confirmation of these theoretical model studies in vivo is still pending [16].

Bioelectrical impedance analysis (BIA) is commonly used to assess body composition based on the electrical characteristics of biological tissue by sending a weak electrical current through the body [17]. One property of electric current is that depending on its frequency, it flows along highly conductive body tissues [18]. While low current frequencies cannot pass through the cells, they flow through the GS [19]. Higher current frequencies, on the other hand, pass through cell membranes, which act as a kind of capacitor and can store and release electrons in a frequency-dependent manner. Therefore, the multi-frequency current used by BIA passes through both the cells and the GS [20]. Most conductive zones in the human body are those with a high percentage of isotonic water (such as the GS) because of its low electrical resistance. The combination of body water and cells creates an impedance to the ability of electric current to flow through them and is referred to as bioimpedance [21]. BIA has gained popularity in tissue research because the devices are noninvasive, inexpensive, and portable [19]. In recent years, BIA has evolved to include individual body parts [22], muscles [23], skin diseases [24], or even carcinogenic skin lesions [25] in addition to whole body analysis.

In order to analyze defined areas of the myofascial tissue, such as the TLF and the underlying erector spinae muscle, four BIA electrodes are placed directly over the region of interest. Whereas the two outer electrodes send a multi-frequency current through the tissue, the two inner electrodes measure the voltage, whose level depends on the impedance of the myofascial tissue (Figure 1) [19]. In a preliminary study, Dennenmoser et al. [26] were able to show that this measurement principle is able to detect an impedance increase after manual manipulation in the TLF, which correlates with higher sonoelastographic stiffness parameters. The aim of this study was to follow this approach and evaluate the bioelectrical properties of the TLF. The authors are hypothesizing that the BIA is mainly influenced by the amount of GS between the layers within the TLF and between it and the muscle. Previous work has shown that fascia stretched over a period of time may respond with fluid loss and a delayed supercompensation effect, in which the amount of fluid in the tissue is higher than before stretching [27]. Therefore, manual manipulation, especially instrument-assisted, as frequently used in manual therapy practice, could likewise produce such an effect. To the authors’ knowledge, this is the first study to investigate the influence of instrument-assisted treatment of TLF on the bioimpedance and hydration content of lumbar myofascial tissue.

## 2. Materials and Methods

The study was a quasi-experiment with one intervention group. Measurements were taken before and after the intervention and at a 3, 5, and 10-min follow-up according to the SPIRIT guidelines [28]. The study protocol was prospectively registered with the German Clinical Trials Register (DRKS00025432) on 17 June 2021. The study has been reviewed and approved by the ethical committee of the DIPLOMA Hochschule, Germany (Nr. 1006/2021), has been carried out in accordance with the declaration of Helsinki and has obtained written informed consent from the participants [29].

### 2.1. Setting and Participants

The study was conducted in a physiotherapy practice in a medium-sized city in northern Germany. The number of participants was calculated with GPower [30] based on the results of a previous study and set at 18 (Cohen’s d = 0.74, α err = 0.05, 1–β err = 0.9) [26]. The acquisition was carried out via direct contact and the distribution of information material in the practice.

Inclusion criteria were: (a) a generally healthy constitution; (b) the persons to be treated must have intact lumbar sensitivity and be able to perceive and communicate pain; (c) a BMI between 18 and 33; (d) female or male subjects aged 18 to 75 years; (e) prone position for 15 min must be pain-free for the subjects.

Exclusion criteria were: (a) generally valid contraindications to physiotherapeutic treatments of the lumbar region; (b) rheumatic diseases; (c) taking medication that affects blood circulation, pain or mind; (d) taking muscle relaxants; (e) skin changes (e.g., neurodermatitis, psoriasis, urticaria, decubitus ulcers, hematoma); (f) surgery or other scars in the lumbar region; (g) previous mental illness; (h) surgery in the last three months; (i) acute inflammation; and (j) pregnancy.

### 2.2. Study Flow

First, anthropometric data (age, sex, height, and weight) were collected by the investigator (MS). Before the measurements, the subjects were informed about intervention and the measurements. They lay in a prone position with their arms at the sides of the body and legs parallel to each other. The head was in a neutral position and the face was in a recess in the head section of the therapy table. The patient was undressed enough to access the TLF between Th12 and S1. A physiotherapist with more than 10 years of experience in manual palpation then marked the vertebrae L1 to L5 and a measuring point 2.5 cm paravertebrally on the right side halfway between L2 and L3 (measuring point: MP_1_, Figure 2). An initial measurement was taken before the intervention (t_0_). Following the intervention (t_1_), and after 5 (t_2_) and 10 min (t_3_) each, further outcome measurements were taken.

### 2.3. Intervention

The therapist applied instrument-assisted soft tissue mobilization (IASTM) with a convex metal therapy tool (Fazer 1, Artzt GmbH, Dornburg, Germany) (Figure 3). During this procedure, the tool was pushed paravertebrally in the craniocaudal direction on both sides of the TLF at a speed of 3 cm/s and a force of 10 N. The therapist was trained in this regard by performing the procedure on a sliding surface of a precision scale before commencing the respective treatments each day. Accuracy was considered acceptable when the therapist applied the procedure three times in succession with a maximum tolerance of 10% (±1 N). The contact area of the tool was 50 mm, and the duration of the entire intervention was 3 min.

### 2.4. Outcomes

Bioimpedance analysis (BIA) was performed using the Nutriguard MS 4-electrode multifrequency analyzer (Data Input GmbH, Wedemark, Germany) with a measurement current of 20 μA, 5–50 kHz. One pair of electrodes was placed paravertebrally on the right TLF at the marked L1 position and the other was placed at the L4 position. This allows the device to record the impedance of the largest possible unilateral TLF area [26]. In healthy individuals with no body shape abnormalities, no fluid imbalance, and within a BMI range of 16 to 34 kg/m^2^, BIA provides reliable information on soft tissue impedance [22]. The measurement error of the Nutriguard MS for the measured values was ±1.2 Ω (±2%) with an excellent test–retest ICC > 0.85 and minimal detectable change (MDC) ranging from 2.0 to 3.25 Ω [31].

TLF stiffness was measured on MP_1_ using indentometry (IndentoPro, Chemnitz University of Technology, Germany and Fascia Research Project, Technical University of Munich, Germany) with an indentation depth of 8 mm [32]. The measurement was performed three times with a time delay of 5 s, and an average value was calculated. The values were acceptable if they were within a measurement range of 10%. The intra-rater reliability is reported to be good (ICC = 0.83–0.84), and the MDC value is 1.55 N/mm based on a standard error of the mean of 0.56 N/mm for a penetration depth of 10 mm, which was investigated in a reliability study by Koch and Wilke [32].

Skin temperature was measured on MP_1_ with an infrared thermometer (FT-90, Beurer GmbH, Ulm, Germany). The measurement error of the device for the measured values was ±0.2 °C.

### 2.5. Statistical Analyses

The standard deviation (SD), mean, minimum (Min), maximum (Max), median, 25% quartile (Q1), and 75% quartile (Q3) were determined for all parameters. The outcome variables were not normally distributed as assessed by the Shapiro–Wilk test (*p* < 0.05). A non-parametric Friedman’s rank test for differences between repeated measures was performed. Significant results were analyzed post hoc using the Dunn test. Moreover, a correlation analysis (rs—Spearman’s correlation coefficient) was performed between the main outcomes and variables such as age, height, weight and BMI. The significance level was set at *p* = 0.05.

Microsoft Excel (Microsoft Corporation©, version 2209, 2019, Albuquerque, NM, USA) was used for the descriptive statistics. The inferential statistics were carried out with Statistica, version 13 TIBCO Software Inc., Palo Alto, CA, USA.

## 3. Results

The anthropometric data and baseline characteristics are shown in Table 1. In total, 21 healthy participants (16 females and 5 males) took part in the study (Figure 4).

The comparison of BIA results is shown in Table 2. A statistically significant increase in bioimpedance was observed in subsequent measurements (*p* < 0.001). The lowest mean value was observed at t_0_ measurement (before treatment) and was 58.3 Ω (SD = 11.1 Ω). The highest was observed at t_3_ measurement (10 min after treatment) and was 60.4 Ω (SD = 10.2 Ω). Multiple comparisons revealed statistically significant differences between t_0_ and t_3_ results (Table 2).

The comparison of the results on tissue stiffness is shown in Table 3. No statistically significant differences were observed between the measurement time points t_0_, t_1_, t_2_ and t_3_.

The comparison of temperature results is shown in Table 4. A statistically significant temperature difference was observed in the subsequent measurements (*p* = 0.011). The lowest mean value was observed at the t_0_ measurement (before treatment) and was 36.1 °C (SD = 1.2 °C). The highest was observed at the t_2_ measurement (5 min after treatment) and was 36.6 °C (SD = 0.7 °C). In multiple comparisons, statistically significant differences were found between t_1_ and t_2_ results (Table 4).

Correlations between BIA results, tissue stiffness, temperature, age, height, body weight, and BMI are shown in Table 5. No statistically significant correlations were found between the variables assessed.

## 4. Discussion

To the authors’ knowledge, this study was the first to investigate the effect of an IASTM on the TLF on lumbar myofascial tissue bioimpedance and its hydration content. The results here showed significant changes in the lumbar BIA and temperature but not in TLF stiffness.

The tissue temperature increased significantly from 5 to 10 min after intervention by 0.3 °C (*p* = 0.029) and also exceeded the measurement error of 0.2 °C. This was to be expected, as friction with a manual therapy device on the skin is supposed to increase tissue temperature [33]. Some authors emphasize that IASTM should increase the range of motion and functionality due to improved blood flow and the associated increased tissue temperature. This would lead to pain relief and a positive change in the compressive, tensile and shear forces acting on the tissue [34,35,36]. However, the mechanisms underlying these processes are as yet unclear. Improved microcirculation as a result of treatment, accompanied by altered expression of cytokines in the ground substance, particularly inflammatory mediators and growth factors that promote regeneration and wound healing, could play a crucial role in improving pathologies in myofascial tissue [35].

BIA showed a significant increase in bioimpedance of 2.1 Ω 10 min after treatment compared to baseline (*p* < 0.001). The value also exceeded the measurement error of 1.2 Ω, indicating that this result was not random. The post-intervention increase in bioimpedance can be interpreted as a decrease in water content in myofascial tissue, whose influence on electrical properties is the greatest compared to other parameters [18]. This observation is consistent with previous studies. Dennenmoser et al. [26] and Cordon et al. [37,38] reported an increase in bioimpedance after manual manipulation of the TLF and fascia lata. Schleip et al. [27] demonstrated that a 4% stretch of the porcine TLF significantly decreased the water content in the fascia after stretching and then gradually returned to baseline levels. They also observed over-recovery or supercompensation, in which a higher water content than at baseline is achieved after a sufficient rest period (Figure 5). An increased stretching force of 6% resulted in an even greater dehydration and rehydration process. In this study, IASTM was applied with a force of 10 N and a velocity of 3 cm/s in the craniocaudal direction to the TLF between L1 and L4. The fibers there mainly follow those of the latissimus dorsi muscle, which inserts into the TLF in the mediocaudal direction. There are also fibers in cross-hatched form that cross these fibers in the craniolateral direction. At this level of the vertebra, the TLF is not as dense as in the lower sections toward the sacrum [1]. We hypothesize that tissue water can be more easily squeezed out of the TLF in the treated area, as evidenced by the significant increase in bioimpedance after IASTM in this study.

There was no statistically significant effect of IASTM on TLF stiffness (*p* = 0.84). This was surprising, since other authors have found a relationship between water content and stiffness [26,37,38]. Therefore, it would be expected that stiffness would increase after a decrease in water content in the TLF. Dennenmoser et al. [26] showed correlations between BIA as an indicator of tissue hydration and TLF stiffness using ultrasound elasticity measurement. Other authors found highly significant correlations for the thigh after self-help treatment with a vibration device [37,38]. We suggest that the discriminative power of stiffness measurement, which was applied only in the perpendicular direction, is insufficient to detect stiffness changes due to an 8–10% change in water content in the 680 μm thin TLF [1]. On the other hand, it may indicate that the water content changes occurred mainly in the overlying soft tissues and not in the muscle tissue, while the stiffness values of the indentometry assessment in this study (with 8 mm indentation depth) may have been influenced more severely by the muscle tissue. Future studies should consider the likely limitations of perpendicular stiffness measurements on the skin and also consider methods capable of determining true TLF stiffness, such as shear wave elastography [39] or ultrasonic deformation measurements [40].

Some limitations need to be discussed. First, the study was conducted in a quasi-experimental design that was not blinded or placebo-controlled. The study group was healthy and had a BMI between 20 and 33, so the applicability of the study results to a symptomatic population with low back pain who normally receive IASTM is limited. The results should be understood under the premise of a basic science study.

Second, significant changes in BIA were noted after IASTM, which also exceeded measurement error. However, the pre- and post-test changes were at the lower limit of the MDC range, and the measurement was performed only on one side of the TLF, while both sides were treated. Therefore, no conclusion could be drawn about crossover effects between different TLF sides. In addition, the TLF is known to couple the latissimus dorsi and the contralateral gluteus maximus [1]. Future studies should consider this mechanism and examine both TLF sides to also observe putative differences.

Third, the multi-frequency BIA used in this study measures the bioimpedance of cells and the GS. Porcine TLF consists of up to 70% water in total [27]. Therefore, the authors assume that the significant changes in bioimpedance after treatment of the TLF were mainly caused by water loss in this tissue. However, muscle tissue is also known to have a high water content (e.g., up to 70% in the abdominal muscle) [41]. However, the erector spinae muscle, whose electrical conductivity could possibly have contributed to the measured BIA values, lies below the TLF and has a dense epimysium with likely higher resistance, which could direct the measurement current mainly through the overlying tissue with higher conductivity. Cadaveric studies focusing on the electrical properties of the lumbar tissue to confirm these considerations are needed. Therefore, the results of this study can only be related to the whole myofascial tissue (erector spinae muscle, subcutaneous tissue, and TLF) in addition to the dermis with its more water-resistant properties.

## 5. Conclusions

To the authors’ knowledge, this is the first study to investigate the effect of IASTM on the BIA of the lumbar myofascial tissue. There was a significant increase in bioimpedance 10 min after HILT intervention, which was likely due to a decrease in water content in myofascial lumbar tissue. Future studies should examine both TLF sites and compare the results from healthy individuals as well as a symptomatic population in a randomized control trial design.

## Figures and Tables

**Figure 1 jcm-12-01009-f001:**
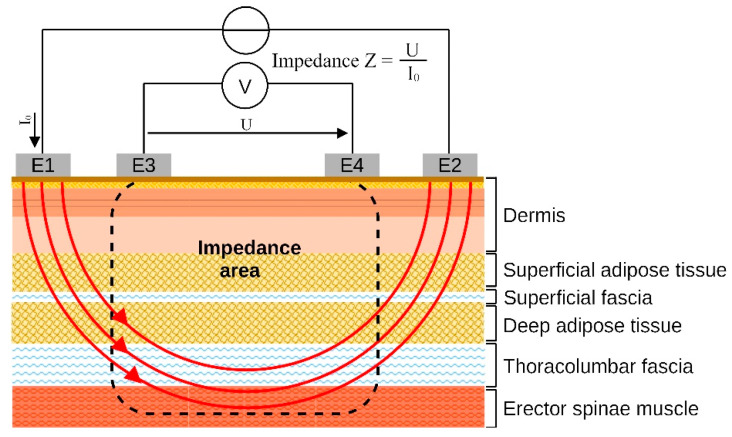
Four electrode method measurement of bioimpedance. E1,2, current electrode; E3,4, voltage electrode; V, voltameter; U, voltage; I_0_, current; Impedance area represents the current-carrying measured range.

**Figure 2 jcm-12-01009-f002:**
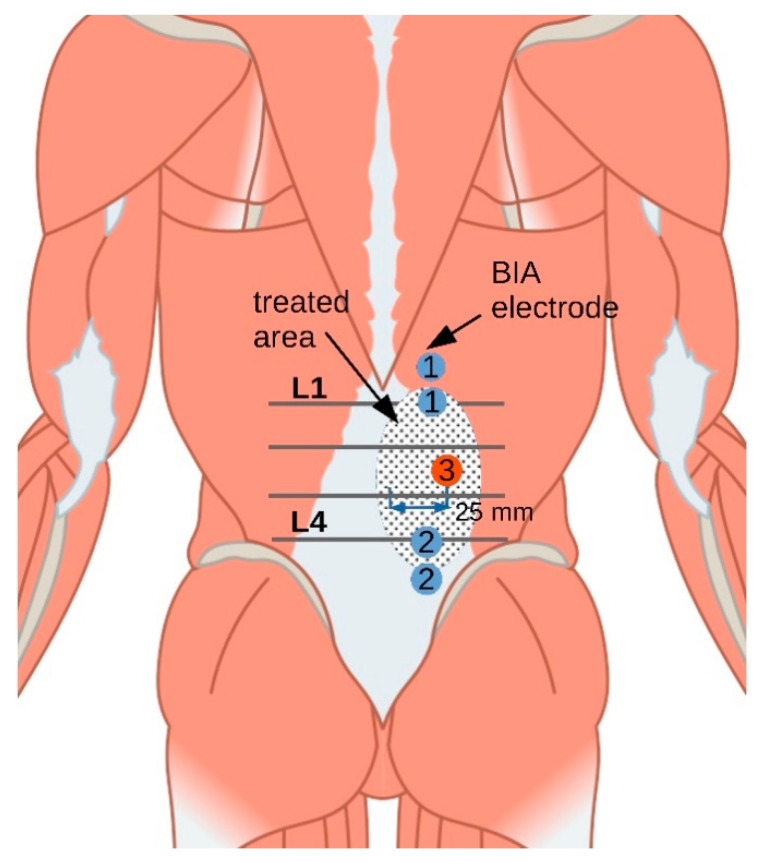
Measuring arrangement. 1, electrode pair 1; 2, electrode pair 2; 3, measurement point MP_1_; BIA, bioimpedance analysis; L1, 1st lumbar vertebra; L4, 4th lumbar vertebra.

**Figure 3 jcm-12-01009-f003:**
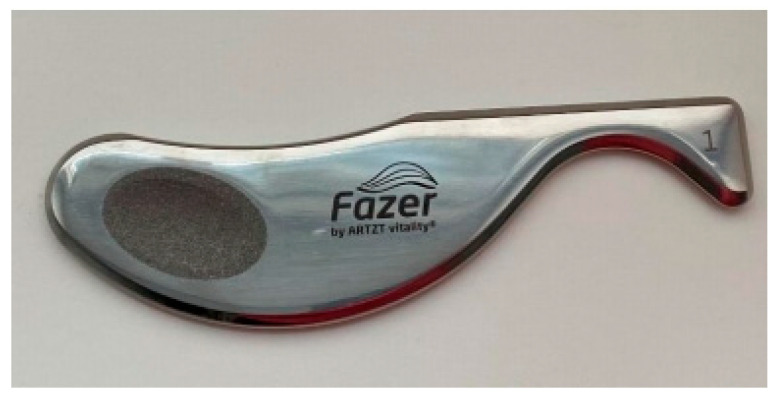
Instrument-assisted soft tissue mobilization tool Fazer 1.

**Figure 4 jcm-12-01009-f004:**
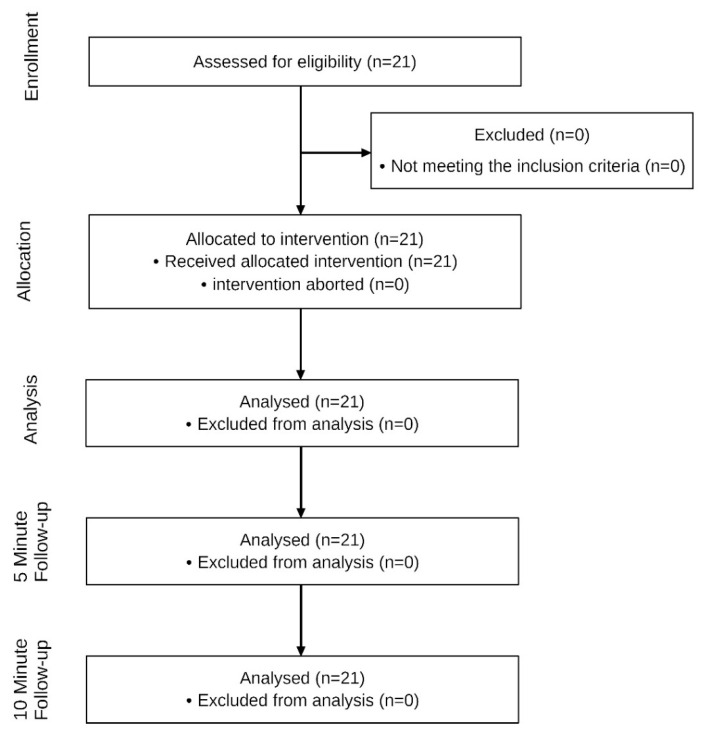
Flow chart of the study.

**Figure 5 jcm-12-01009-f005:**
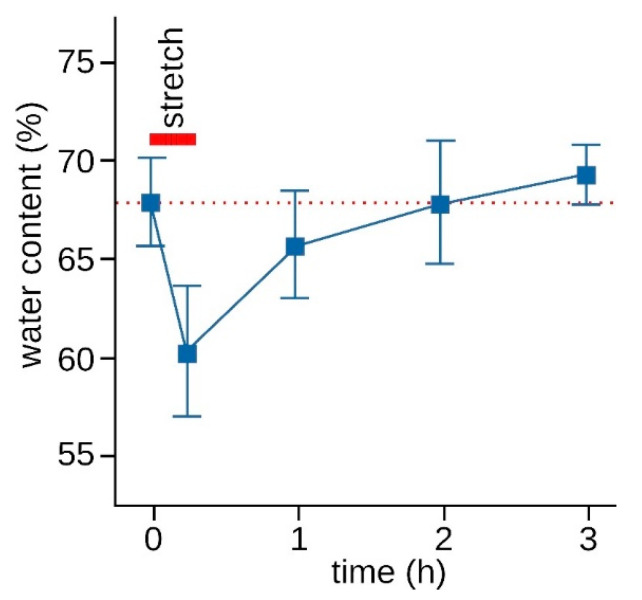
Changes in water content after 15 min of stretching according to Schleip et al. [27]. The figure shows the mean water content of lumbar fascia samples from porcine (*n* = 25) after a 15-min stretch of 4%. The error bars show the standard deviation.

**Table 1 jcm-12-01009-t001:** Baseline characteristics.

Study Group (*n* = 21)							
Quantitative Variable	Descriptive Statistics						
	M	Me	Min	Max	Q1	Q3	SD
Age (years)	41.9	42.0	18.0	75.0	36.0	49.0	14.3
Height (cm)	173.3	172.0	162.0	191.0	166.0	179.0	8.1
Weight (kg)	77.1	76.0	60.0	105.0	65.0	85.0	13.7
Body Mass Index (kg/m^2^)	25.5	25.0	20.0	33.0	24.0	27.0	3.4
Qualitative variable		*n*			%		
Gender	Female	16			76		
	Male	5			24		

*n*, number of participants. M, mean. Me, median. Min, minimum value. Max, maximum value. Q1, 25% quartile. Q3, 75% quartile. SD, standard deviation.

**Table 2 jcm-12-01009-t002:** Bioimpedance analysis.

Study Group (*n* = 21)							
Bioimpedance (Ω)	Descriptive Statistics						
	M	Me	Min	Max	Q1	Q3	SD
t_0_	58.3	57.3	43.0	86.7	48.5	66.0	11.1
t_1_	60.0	58.0	44.3	86.9	52.1	66.0	10.3
t_2_	60.0	58.0	44.8	86.9	53.0	66.0	10.1
t_3_	60.4	58.6	44.9	87.0	53.0	67.0	10.2
*p*-value (main effect) *	*p* < 0.001						
*p*-value (multiple comparisons) **	t0:t1—*p* = 0.20						
	t0:t2—*p* = 0.07						
	t0:t3—*p* < 0.001						
	t1:t2—*p* = 0.99						
	t1:t3—*p* = 0.05						
	t2:t3—*p* = 0.15						

*n*, number of participants. M, mean. Me, median. Min, minimum value. Max, maximum value. Q1, 25% quartile. Q3, 75% quartile. SD, standard deviation. t_0_, before treatment. t_1_, immediately after treatment. t_2_, 5 min after treatment. t_3_, 10 min after treatment. * Friedman’s rank test. ** Dunn’s test.

**Table 3 jcm-12-01009-t003:** Stiffness of the thoracolumbar fascia.

Study Group (*n* = 21)							
Tissue Stiffness (N/mm)	Descriptive Statistics						
	M	Me	Min	Max	Q1	Q3	SD
t_0_	0.69	0.71	0.13	1.40	0.41	0.89	0.38
t_1_	0.68	0.62	0.13	1.44	0.39	0.91	0.41
t_2_	0.67	0.56	0.12	1.43	0.37	0.93	0.39
t_3_	0.65	0.55	0.13	1.52	0.33	0.94	0.38
*p*-value (main effect) *	*p* = 0.84						
*p*-value (multiple comparisons) **	-						

*n*, number of participants. M, mean. Me, median. Min, minimum value. Max, maximum value. Q1, 25% quartile. Q3, 75% quartile. SD, standard deviation. t_0_, before treatment. t_1_, immediately after treatment. t_2_, 5 min after treatment. t_3_, 10 min after treatment. * Friedman’s rank test. ** Dunn’s test.

**Table 4 jcm-12-01009-t004:** Temperature analysis.

Study Group (*n* = 21)							
Temperature (Celsius)	Descriptive Statistics						
	M	Me	Min	Max	Q1	Q3	SD
t_0_	36.1	36.5	34.2	38.4	34.6	36.6	1.2
t_1_	36.3	36.4	34.0	37.8	36.2	36.7	0.8
t_2_	36.6	36.6	34.4	38.0	36.3	36.9	0.7
t_3_	36.4	36.6	34.2	38.0	36.2	36.9	1.0
*p*-value (main effect) *	*p* = 0.011						
*p*-value (multiple comparisons) **	t0:t1—*p* = 0.99						
	t0:t2—*p* = 0.06						
	t0:t3—*p* = 0.58						
	t1:t2—*p* = 0.029						
	t1:t3—*p* = 0.41						
	t2:t3—*p* = 0.86						

*n*, number of participants. M, mean. Me, median. Min, minimum value. Max, maximum value. Q1, 25% quartile. Q3, 75% quartile. SD, standard deviation. t_0_, before treatment. t_1_, immediately after treatment. t_2_, 5 min after treatment. t_3_, 10 min after treatment. * Friedman’s rank test. ** Dunn’s test.

**Table 5 jcm-12-01009-t005:** Correlations between variables.

Variable	Timepoint	Bioimpedance (Ω)		Tissue Stiffness (N/mm)		Temperature (Celsius)	
		r_s_	*p*-Value	r_s_	*p*-Value	r_s_	*p*-Value
Age	t_0_	0.14	0.54	0.08	0.75	−0.23	0.31
	t_1_	0.09	0.68	0.04	0.88	−0.19	0.41
	t_2_	0.12	0.59	0.14	0.56	−0.12	0.59
	t_3_	0.12	0.61	0.08	0.75	−0.14	0.56
Height (cm)	t_0_	−0.18	0.43	0.21	0.36	0.09	0.70
	t_1_	−0.30	0.18	0.16	0.49	0.07	0.76
	t_2_	−0.34	0.13	0.24	0.30	−0.04	0.87
	t_3_	−0.34	0.13	0.31	0.18	−0.12	0.60
Weight (kg)	t_0_	0.08	0.74	0.07	0.77	−0.17	0.47
	t_1_	0.05	0.83	−0.01	0.96	0.12	0.59
	t_2_	−0.01	0.96	0.07	0.77	0.05	0.84
	t_3_	−0.01	0.96	0.11	0.64	0.03	0.89
Body Mass Index (kg/m^2^)	t_0_	0.16	0.50	0.02	0.91	−0.27	0.23
	t_1_	0.20	0.39	−0.05	0.82	0.16	0.50
	t_2_	0.15	0.52	−0.01	0.97	0.12	0.60
	t_3_	0.15	0.53	0.02	0.95	0.16	0.49

r_s_, Spearman’s correlation coefficient. t_0_, before treatment. t_1_, immediately after treatment. t_2_, 5 min after treatment. t_3_, 10 min after treatment.

## Data Availability

Data can be made available by the author upon request.

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
