# Peer review of "Immediate Effects of Instrument-Assisted Soft Tissue Mobilization on Hydration Content in Lumbar Myofascial Tissues: A Quasi-Experiment"

_jcm, 2023, doi:10.3390/jcm12031009_

Round 1

Reviewer 1 Report

Dear Author

1) How calculate the sample size?

2)Please calculate the reliability(ICC,SEM,MDC) of the MS and indentometry?

3) Please revise the results and discussion based on new analysis.

Author Response

Dear Reviewer,

We were very impressed with the thorough review and the many valuable recommendations and improvements we received from you. We are convinced that this will significantly improve the quality of our article. We have revised our article in this regard, following your recommendations step by step. Below we comment point by point on your comments.

  1. How calculate the sample size?
    We calculated the sample size based on the effect size of a previous intervention study using a similar treatment. We added the data on which our calculation was based in this context. Lines 114-116.
  2. Please calculate the reliability(ICC,SEM,MDC) of the MS and indentometry?
    ICC, SEM, and MDC should be results of a carefully designed reliability study in which a sufficient number of raters take measurements on the same sample under identical conditions. This was not the focus of our study. Therefore, we reviewed the reliability data for our measurement instruments from previous reliability studies and presented them in the methodology section, which is a common practice in intervention study design. To further highlight this, we added ICC and MDC for the myofascial BIA analysis and mentioned our findings in this regard in the limitations section. Lines 163-165; lines 293-294.
  3. Please revise the results and discussion based on new analysis.
    We followed the recommendation and revised our manuscript accordingly (see items 1 and 2).

We have explained your concerns in detail and have revised our paper according to your recommendations and have made efforts to improve our manuscript. 

Reviewer 2 Report

Dear Editor

I would like to thank you for the opportunity to review the manuscript entitled "Immediate effects of instrument-assisted soft tissue mobilization on hydration content in lumbar myofascial tissues. A Quasi-Experiment".

The aim of this study was carried out with a well-structured methodology, the writing is clear, and the study’s limitations are presented and discussed. Moreover, the authors reinforced the importance of performing randomized control trial design studies to extrapolate the results in healthy subjects to a symptomatic population as well as to confirm the reliability of BIA in the myofascial tissue. 

Author Response

Dear Reviewer,

we were very impressed by the thorough review and your valuable comments, which take into account our great efforts to produce our manuscript objectively and with high quality.

Reviewer 3 Report

Dear Authors,

the paper is well written and original, but the results can't support the discussion. Indeed it isn't clear how you have standardized the manual treament. Besides, all the techniques are not specific for the TLF, but they could be affected also by changes in the skin, in the adipose tissues... Besides, the indentometer has a validation for a penetration depth of 10 mm, but often the subcutaneous tissue in the lumbar area is thicker. 

Author Response

Dear Reviewer,

We were very impressed with the thorough review and the many valuable recommendations and improvements we received from you. We are convinced that this will significantly improve the quality of our article. We have revised our article with this in mind and have followed your recommendations step by step. In the following, we address your comments point by point.

  1. The paper is well written and original, but the results can't support the discussion.
    We have discussed in detail, point by point, the concerns about the validity of the results and whether they can be discussed in this way, and we have supported our arguments with references (see items 2-4).
  2. Indeed it isn't clear how you have standardized the manual treament.
    We described the standardization of the intervention in a separate subsection and in a detailed figure within the methodology section. Figure 2; item 2.3; lines 143-154.
  3. Besides, all the techniques are not specific for the TLF, but they could be affected also by changes in the skin, in the adipose tissues.
    Because the TLF is an aponeurotic fascial structure located between the skin, subcutaneous adipose tissue (SAT), and erector spinae muscle, there is currently no manual treatment technique whose results could be considered exclusively for the TLF. To focus more on the myofascial tissue, we used indentometry with a higher penetration depth (see next discussion point) and BIA, which makes reliable measurements[1],[2],[3] in this tissue rather than in the skin or SAT, where electrical resistance is 102 higher[4]. We also made it clear that the results of our study can only make a statement for the entire myofascial tissue in the lumbar region and must be seen as a basic science study. Lines 287-310.
  4. Besides, the indentometer has a validation for a penetration depth of 10 mm, but often the subcutaneous tissue in the lumbar area is thicker.
    Koch and Wilke[5] examined the IndentoPro device in a validity and reliability study. The results showed that the skin and the SAT had no influence on the stiffness measurement with the IndentoPro device, even at a low penetration depth of 5 mm. However, you are correct in that it is questionable whether identometry, as well as other perpendicular methods on the skin, are able to detect changes in the thin TLF, which the most recent reliability study suggests[6]. Therefore, we urged in the Discussion section that this be considered in future studies. Lines 283-286.

We have presented your concerns in detail and revised our article according to your recommendations and have made efforts to improve our manuscript. 

[1] Bertuccioli, A.; Cardinali, M.; Benelli, P. Segmental Bioimpedance Analysis as a Predictor of Injury and Performance Status in Professional Basketball Players: A New Application Potential? Life 2022, 12, 1062, doi:10.3390/life12071062.

[2] Das, L.; Das, S.; Chatterjee, J. Electrical Bioimpedance Analysis: A New Method in Cervical Cancer Screening. J Med Eng 2015, 2015, e636075, doi:10.1155/2015/636075.

[3] Dennenmoser, S.; Schleip, R.; Klingler, W. Clinical Mechanistic Research: Manual and Movement Therapy Directed at Fascia Electrical Impedance and Sonoelastography as a Tool for the Examination of Changes in Lumbar Fascia after Tissue Manipulation. J Bodyw Mov Ther 2016, 20, 145, doi:10.1016/j.jbmt.2015.07.021.

[4] 1.  Faes, T.J.C.; Meij, H.A. van der; Munck, J.C. de; Heethaar, R.M. The Electric Resistivity of Human Tissues (100 Hz-10 MHz): A Meta-Analysis of Review Studies. Physiol. Meas. 1999, 20, R1–R10, doi:10.1088/0967-3334/20/4/201.

[5] 1.  Koch, V.; Wilke, J. Reliability of a New Indentometer Device for Measuring Myofascial Tissue Stiffness. J Clin Med 2022, 11, 5194, doi:10.3390/jcm11175194.

[6] 1.  Bartsch, K.; Brandl, A.; Weber, P.; Wilke, J.; Bensamoun, S.F.; Bauermeister, W.; Klingler, W.; Schleip, R. The Princess and the Pea - Comparison of Different Stiffness Assessment Tools on a Multi-Layered Phantom Tissue Model. Research Square 2022, doi:10.21203/rs.3.rs-2039032/v1.

Round 2

Reviewer 3 Report

thank you for the revised version and to have answer to all the questions